# Long-Term Outcomes of Contemporary Percutaneous Coronary Intervention with the Xience Drug-Eluting Stent: Results from a Multicentre Australian Registry

**DOI:** 10.3390/jcm12010280

**Published:** 2022-12-29

**Authors:** David S. Eccleston, Enayet Chowdhury, Tony Rafter, Peter Sage, Alan Whelan, Christopher Reid, Danny Liew, MyNgan Duong, Nisha Schwarz, Stephen G. Worthley

**Affiliations:** 1Melbourne Private Hospital, Melbourne, VIC 3052, Australia; 2GenesisCare Cardiology, GenesisCare, Leabrook, Adelaide, SA 5068, Australia; 3Wesley Hospital, Auchenflower, Brisbane, QLD 4066, Australia; 4St Andrews Medical Clinic, Adelaide, SA 5000, Australia; 5Wexford Medical Centre, Murdoch, Perth, WA 6150, Australia; 6Faculty of Health Sciences, Curtin University, Perth, WA 6845, Australia; 7Adelaide Medical School, The University of Adelaide, Adelaide, SA 5000, Australia; 8North Shore Cardiology, St Leonards, Sydney, NSW 2065, Australia

**Keywords:** percutaneous coronary intervention, outcomes, Xience, drug-eluting stent, bare metal stent, registry

## Abstract

**Introduction:** Several large registries have evaluated outcomes after percutaneous coronary intervention (PCI) in the USA, however there are no contemporary data regarding long-term outcomes after PCI, particularly comparing new generation drug-eluting stents (DES) with other stents in Australia. Additionally, approval of new-generation drug-eluting stents (DES) is almost exclusively based on non-inferiority trials comparing outcomes with early generation DES, and there are limited data comparing safety and efficacy outcomes of new-generation DES with bare metal stents (BMS). This study reports in-hospital and long-term outcomes after PCI with the Xience DES from a large national registry, the GenesisCare Outcomes Registry (GCOR). **Methods:** The first 1500 patients consecutively enrolled from January 2015 to January 2019 and treated exclusively with either Xience DES or BMS and eligible for 1-year follow-up were included. Baseline patient and procedural data, major adverse cardiovascular events (MACE) in-hospital, at 30 days and 1-year, and medications were reported and analysed with respect to Xience DES (*n* = 1000) or BMS (*n* = 500) use. **Results:** In this cohort the mean age was 68.4 ± 10.7 years, 76.9% were male, 24.6% had diabetes mellitus and 45.9% presented with acute coronary syndromes. Of the overall cohort of 4765 patients from this period including all DES types, and patients who received multiple DES or a combination of DES and BMS, DES were exclusively used in 3621 (76.0%) patients, and BMS were exclusively used in 596 (12.5%). In comparison to international cohorts, adverse clinical event rates were low at 30 days in terms of mortality (0.20%), target lesion revascularisation (TLR, 0.27%) and MACE (0.47%), and at 12 months for mortality (1.26%) TLR (1.16%) and MACE (1.78%). **Conclusions:** Clinical practice and long-term outcomes of PCI with the Xience DES in Australia are consistent with international series. Recent trends indicate DES use has increased in parallel with good outcomes despite an increasingly complex patient and lesion cohort.

## 1. Introduction

Percutaneous coronary intervention (PCI) is one of the commonest procedures in cardiology, with rapidly evolving technology and clear guidelines from large-scale randomised clinical trials for post-procedural management [1,2]. Modern Drug-eluting stents (DES) perform better than bare metal stents (BMS) and first-generation DES with respect to in-stent restenosis (ISR) and stent thrombosis (ST) and are currently preferred in PCI for most indications. Despite improvements, ISR, and to a lesser extent periprocedural myocardial infarction (MI) and ST are still relevant clinical issues. One recent study reported clinically driven repeat revascularisation in 16.5% of DES treated patients [3]. The Xience everolimus eluting DES (Abbott Laboratories, Abbott Park, IL, USA), introduced in 2006, is one of the most commonly used DES worldwide. Despite this, there are no national data in Australia regarding outcomes after PCI or compliance with medication guidelines [4,5]. Drug-eluting stents (DES) have become the cornerstone of PCI practice after studies indicating superiority to bare metal stents (BMS) [6]. However, DES use has expanded greatly both in terms of patient and lesion characteristics and clinical presentation to include use in very long lesions, small vessels, restenosis and ST-elevation myocardial infarction (STEMI). Additionally, approval of new-generation drug-eluting stents (DES) has been based almost exclusively on non-inferiority trials comparing outcomes with early-generation DES and have typically shown similar efficacy and similar or superior safety. However, there are limited data comparing safety and efficacy outcomes of new-generation DES with BMS, and the assumption that the performance of all new-generation DES is superior to bare-metal stents is less clear [7]. As such, it is timely to compare contemporary outcomes after PCI using DES and specifically newer platforms such as Xience DES with BMS in a real-world cohort to guide optimal clinical practice. Registries offer an important supplementary source of data to inform evidence-based practice as they reflect real world conditions [1,2,3,4,5,6,7,8,9].

The GenesisCare Cardiovascular Outcomes Registry (GCOR) is a multi-centre national observational registry that was established to improve the quality of cardiovascular disease management in Australia by reporting and benchmarking local performance and compliance with treatment guidelines against local and international outcomes [10]. GCOR provides a national snapshot and real-time continuous outcomes measures that are not available elsewhere for one of the commonest and most critical cardiovascular procedures performed in Australia.

The GCOR-Percutaneous Coronary Intervention Registry (GCOR-PCI) commenced in 2008 and has collected detailed clinical and procedural data on more than 14,000 PCI procedures performed in Australian private hospitals. Nearly 50% of all PCIs in Australia are performed in private hospitals [1,11]. Overseas data indicate DES are used preferentially in most PCI settings, and an increasing proportion of patients undergoing emergent or semi-urgent PCI [12], however despite an increasing burden of disease and healthcare expenditure, there are no contemporary national data regarding the demographic, procedural and long-term outcomes after PCI with Xience DES in Australia. The aim of this observational study was to analyse the characteristics and outcomes of PCI with Xience DES in Australian patients using GCOR Registry data. Additionally, we aimed to compare the clinical indications for, and outcomes of Xience DES compared to BMS.

## 2. Methods

This observational cohort study included consecutive patients from the GCOR registry. GCOR-PCI is a prospective clinical quality registry that describes the contemporary management, in-hospital and long-term outcomes of patients undergoing PCI at private hospitals across Australia, while providing continuous performance and outcome measure feedback to participating hospitals and cardiologists and has been previously described [10]. As such, it is more comprehensive and has greater external validity than registries recruiting from single centres or only public sector centres in a single city, such as Melbourne Interventional Group, and which provide no direct outcome feedback to interventionalists [8]. Briefly, data are entered on consecutive patients using a web-based electronic case report form (eCRF). The Registry employs data elements that are aligned with the US National Cardiovascular Data Registry (NCDR), allowing comparisons with international cohorts. The registry is housed at the Centre for Clinical Research Excellence in Therapeutics (CCRET, Monash University, Commercial Road, Prahran). CCRET meets standards relating to the use of paperless records under the Good Clinical Practice regulations and complies with the National E-Health Transition Authority (NEHTA) standard of reporting and storing data. The systems and processes with respect to privacy and data protection comply with relevant Health Records and Information Privacy Acts and Information Privacy Principles. This study is included in the Australian New Zealand Clinical Trials Registry (ANZCTR ACTRN12620000899943). Funding is provided internally to avoid the potential for bias from industry or governmental sources and none of the authors have competing interests [10]. 

Inclusion criteria were the first 1500 consecutively enrolled patients in the GCOR registry undergoing PCI for acute coronary syndromes or electively for symptomatic ischaemic heart disease using either exclusively BMS or Xience DES (Abbott Laboratories, Abbott Park, IL, USA) between January 2015 and January 2019. Exclusion criteria were use of any non-BMS stent other than Xience or combined use of BMS, Xience and other DES in one procedure.

Procedure details and In-hospital complications were recorded at the time of hospital discharge. Thirty day and 12-month follow-up is performed by research coordinators at each hospital [13,14]. All cardiac events were documented following review of medical records including death, myocardial infarction (MI), target lesion revascularisation (TLR, defined as revascularisation within 5 mm of a previously treated segment), and target vessel revascularisation (TVR, defined as revascularisation of a previously treated artery). Composite major adverse cardiovascular events (MACE, comprising death, MI, and TVR) were reported. Enrolling sites and 5% of patient data are independently audited each year [15,16]. Over 99% of patients enrolled are followed beyond one year, with continuing follow-up for the entire dataset to 5 years.

Clinical and procedural management was determined by the treating cardiologist with regard to currently accepted guidelines. Stent diameter was used as a surrogate for target vessel diameter, and total stent length for lesion length. 

Each centre received written Human Research Ethics Committee (HREC) approval for collection of patient data and follow-up prior to participation (Bellberry HREC, Eastwood, SA, Australia); stored data is de-identified. An opt-out consent process was employed following the example of previous models such as Melbourne Interventional Group (MIG) [8]. This strategy has been effective in achieving high rates of participation essential to an effective registry, and in minimising bias such as the “Hawthorne” effect. 

We compared patients according to stent type and acuity. Patients were included who received exclusively Xience DES or BMS. We carried out several analyses using logistic regression to determine the association between baseline clinical and procedural variables and adverse clinical outcomes at 30 days and 1-year. Firstly, we carried out unadjusted analysis for type of stent used (Xience or BMS). Secondly, we undertook an analysis adjusted for age, gender, previous history of hypercholesterolaemia, family history, MI, PVD, PCI, CVD, CABG and BMI. Continuous variables were summarised as means and SDs and compared using Student *t* tests. Categorical data were summarised as percentages and compared using X^2^ tests. Two-sided *p* values of < 0.05 were considered statistically significant. All analyses were carried out using statistical software Stata version 14.1 (StataCorp LP, College Station, TX, USA).

## 3. Results

The GCOR-PCI Xience/BMS cohort comprised 1500 coronary interventions. Patients were predominantly male (77.3%), with a mean age of 68.4 ± 10.7 years (Table 1). 

In over half (51.2%) of the 1500 procedures the indication for PCI was presentation with an acute coronary syndrome. Of those, 9.6% had ST-segment elevation MI (STEMI), 24.2% non-STEMI and 16.4% unstable angina. Most lesions were de novo (95.0%), and almost half of the cohort had complex lesions (46.6% type B2/C) (Table 2). The mean stent length was 18.1 mm (SD, 5.7 mm) and the mean stent diameter was 3.1 mm (SD, 0.5 mm). Drug-eluting stents (DES) were used exclusively in 76.0% and BMS exclusively in 12% of the overall cohort, with the remainder using a combination of BMS and DES.

Procedural success rates were high (99%) and complication rates very low, consistent with international series. In-hospital mortality was 0.40%, and MI occurred in 2.9%. (Table 3).

At 30-day follow-up of 1494 patients (99.6% of those eligible), rates of unplanned readmission (1.7%) and clinical events such as mortality (0.22%), Mi (0.0%), TVR (0.27%), TLR 0.27%, and overall MACE (0.47%) were low.

At 12 months, follow-up of 1491 patients (96.8% of those eligible) revealed an unplanned readmission rate of 7.4%, with similarly low rates of mortality (1.69%), MI (0.61%), TVR (0.95%), TLR (1.08%) and MACE (3.18%).

Data from a total of 1950 lesions treated during 1500 procedures, in patients who received exclusively either Xience DES or BMS, were used for comparison of patient and lesion characteristics and outcomes between those receiving Xience DES and BMS. Of these, Xience DES were used in 1375 (66.7%) and BMS in 585 (33.3%) lesions. Multiple coronary lesions were more commonly treated with Xience ES than BMS (1.41 ± 0.7 vs. 1.18 ± 0.5 lesion per procedure, *p* < 0.001).

Patients receiving Xience DES were younger than those receiving BMS (66.9 ± 10.1 vs. 71.5 ± 11.1 years, *p* <0.001) however otherwise had more high-risk characteristics including diabetes (26.2% vs. 21.5%, *p* = 0.05), hypercholesterolaemia (85.6% vs. 79.4%, *p* < 0.001), previous MI (23.9% vs. 16.6%, *p* = 0.005) and prior PCI (34.7% vs. 17.1%, *p* < 0.001), however were less likely to have a history of peripheral vascular disease (7.0% vs. 10.0%, *p* = 0.048) or cerebrovascular disease (6.3% vs. 10.2%, *p* = 0.008). However, patients receiving Xience DES were less likely to present with acute coronary syndromes than those receiving BMS (42.9% vs. 63.6%, *p* < 0.001), particularly in the setting of STEMI (4.3% vs. 20.0%, *p* < 0.001) or cardiogenic shock (0.2% vs. 1.0%, *p* <0.001). Overall, Xience DES were more often used in elective than acute presentations (56.6% vs. 43.4% *p* < 0.001) whereas BMS were far more likely to be used in acute than elective presentations (63.6% vs. 36.4% *p* < 0.001).

Patients with complex lesion morphology were more likely to receive Xience DES than BMS, consistent with clinical trial data, Xience DES use was greater for In-stent restenosis (5.5% vs. 1.4%, *p* = 0.002), LAD lesions (47.1% vs. 32.3%, *p* = 0.003), ACC/AHA B2 or C morphology (49.9% vs. 40.0% *p* < 0.001), bifurcations (11.5% vs. 4.8%, *p* < 0.001), long lesions >20 mm (30.2% vs. 17.0%, *p* <0.001), and small vessels <2.5 mm diameter 25.2% vs. 12.8%, *p* < 0.001) (Table 2).

There was a trend towards higher in-hospital mortality in the BMS than the Xience DES cohort although this was not statistically significant (0.8% vs. 0.2%; *p* = 0.08), and there was no difference in the rate of major bleeding between groups (BARC 3 or 5 1.4% vs. 1.8%; *p* = 0.56). Despite Xience DES patients having more high-risk patient and lesion characteristics, they had lower rates of 30-day mortality (0.0% vs. 0.6%; *p* < 0.001) and MACE (0.2% vs. 1.0%; *p* <0.001) than patients receiving BMS (Table 3). 

At 12-month follow-up, the association of lower mortality with Xience DES use became more evident, where Xience DES patients had a 70% lower mortality compared to those with BMS (1.1% vs. 3.5%, *p* 0.002 adjusted OR 0.70 (0.10–0.91, *p* < 0.03). MACE rates were similar between groups 2.9% vs. 4.5% OR 0.70, *p* = 0.12, adjusted OR 0.70 *p* = 0.35), (Table 4). Myocardial infarction at 30 days and at 12 months was uncommon and late stent thrombosis rare; rates were similar in both cohorts. All these analyses are adjusted for possible confounders. Further adjustment with propensity scoring as a continuous variable reveals similar findings [17,18,19,20]. 

### 3.1. Case Acuity and Outcomes

Overall, 747 of 1500 of patients (50%) presented with acute coronary syndromes, with 143 (9.5%) presenting with STEMI, 360 (24%) with NSTEMI and 244 (16.2%) with unstable angina (Appendix A). Patients presenting with ACS were younger than those undergoing elective PCI for both BMS (71.1 years vs. 72.2 years, *p* < 0.001) and Xience (66.2 years vs. 67.4 years, *p* < 0.001) groups. Patients presenting with STEMI more commonly received BMS than Xience (31.4% vs. 10.0%, *p* < 0.001), whereas those presenting with NSTEACS were more likely to receive Xience DES than BMS (56.7% vs. 43.2%, *p* <0.01)

At 12 months risk-adjusted mortality in patients undergoing elective PCI was low, and lower in those receiving Xience DES than BMS (0,5% vs. 2.8% *p* = 0.01, adjusted OR 0.16 (0.03–0.92, *p* = 0.04) (Table 4). In patients presenting with ACS there was a non-significant trend for lower mortality in those receiving Xience compared to BMS (1.9% vs. 3.9% *p* = 0.11) MACE at 12 months was lower in elective patients receiving Xience than BMS stents (1.8% vs. 4.9%, *p* = 0.04, however this was not significant after adjustment for elective or ACS patients. 

### 3.2. Medication Compliance

Compliance with guideline-recommended secondary prevention medications was consistently high in the overall cohort and both stent groups at discharge (aspirin 98.3%, clopidogrel, ticagrelor or Prasugrel 98.5%, statin 94.6%). (Table 5) Compliance remained high at 12 months (antiplatelet therapy 93.3%, statin 94.7%). Patients with DES were more likely to receive dual anti-platelet therapy than patients with BMS at both 30 days (97.8% vs. 91.6%; *p* < 0.001) and 12 months (74.9% vs. 43.7%; *p* < 0.001). The rates of use of β-blockers and angiotensin-converting enzyme (ACE) inhibitors were lower than expected at both 30 days and 12 months.

## 4. Discussion

This study represents the first registry of national scale reporting outcomes following PCI with Xience DES in Australia. It provides a unique perspective on contemporary practice and outcomes of PCI in Australia, and a large-scale report regarding PCI performance in Australian private hospitals. Despite some perceptions that physicians in private hospitals perform mainly elective procedures, nearly half (46%) of all patients presented with acute coronary syndromes. Similarly, a large proportion had complex lesion types (e.g., type B2/C).

Overall, patient and procedural characteristics in this study were similar to reports from regional public hospital and international reports such as the Melbourne Interventional Group (MIG), Swedish Web-system for Enhancement and Development of Evidence-based care in Heart disease Evaluated According to Recommended Therapies (SWEDEHEART) and NCDR Cath-PCI [1,8,9,21,22,23,24]. In this Xience DES study the majority of patients were treated exclusively with DES (76%), consistent with recent international series [25,26,27,28]. However, whilst the patients and lesions treated with DES in this study had more high-risk characteristics than those in early trials of DES vs. BMS, outcomes with the Xience DES remained consistently superior to BMS in line with the benefits shown in studies of earlier generation DES vs. BMS [6,9,17,18,19,20].

However, in this cohort 30-day unplanned readmission rates were significantly lower than in NCDR Cath-PCI (1.7% vs. 11.9%, *p* < 0.001) [25]. Rates of 12-month target vessel revascularisation were relatively low in comparison with registries such as MIG (0.9% vs. 6.9% overall), and mortality rates were significantly lower than in the MIG cohort at 30 days (0.20% vs. 1.9%, *p* = 0.0001) and 12 months (1.7% vs. 5.2%, *p* = 0.0001) [25]. These favourable outcomes are consistent with recent data supporting the increased long-term safety and low late thrombosis rates of newer generations of DES such as Xience [27,28]. 

By comparison a recent report of a large European cohort found the overall rates of ISR and ST were similarly low. Additionally, with stents stratified in small, medium, and large sizes, angiographic outcomes were similar for Xience compared to other DES. Stent and lesion characteristics were similar in Xience and other DES with mean stent diameter 3.0 mm and length 21 mm. Crude rates of ISR in Xience were 2.9% and ST were 0.9%, respectively in the SCAAR cohort, similar to the GCOR group [29].

We have previously reported excellent and continuously improving compliance with guideline pharmacotherapies post-PCI in the GCOR Registry, in part due to continuous quality improvement pathways facilitated by a broad network approach to patient care [9]. In this study group medication compliance was similarly maintained between 30 days and 12-month follow-up. Dual Antiplatelet therapy (DAPT) was maintained in the majority (67.7%) at 12 months, although this was more common in Xience DES than BMS patients (74.9% vs. 43.7% *p* < 0.001), reflecting recently updated American Heart Association/American College of Cardiology guidelines recommending that DAPT can be ceased in elective patients after 1 month with BMS and 6 months in those receiving newer generation DES such as Xience [9,30,31].

Potential limitations of this study are the fact that while this is a large multi-centre study, involving urban major teaching hospitals and regional centres, only private Australian hospitals were included. However, given existing data collection limitations, this is the best information of national scope that is available to represent contemporary PCI practice and outcomes in Australia. Whilst patients treated in private hospitals might potentially be considered to differ from those managed in the public sector our data demonstrated patient and lesion characteristics comparable to those in MIG, a large Victorian Public Hospital Registry, and international series such as NCDR Cath-PCI [9,25,28]. Patients managed in other private hospitals may receive different care from those in public hospitals, and hence these data may not reflect practice and outcomes in hospitals not participating in this registry.

However, notwithstanding these factors, this is a large contemporary clinical quality registry of PCI practice in Australia which can help health care providers determine the effectiveness of translation of evidence-based guidelines into real-world practice. Importantly, registries are also a key post-marketing surveillance, which serves to protect patients by providing some form of accountability measure following approval by various bodies of either a drug or therapeutic device (DES being a prime example), which is often based on results of randomised controlled trials (RCT) with inherent limitations of size and the inability to monitor for lower frequency yet high significance events such as stent thrombosis. secondary prevention compliance or long-term health, quality of life and cost-effectiveness outcomes that are critical in healthcare [21,22,29,32]. We are continuing follow-up to 5 years in all patients to maximise the value of the system to detect late events, provide quality assurance and to continue to improve patient compliance with appropriate medical therapy. 

## 5. Conclusions

The GCOR Xience DES Registry is the first Australian registry to report contemporary PCI practice and outcomes with Xience DES compared to BMS in Australia. Overall clinical event rates were consistent with those from international reports, with lower 30-day rates of readmission. Despite greater lesion complexity in patients receiving Xience DES, outcomes were superior to those in patients treated with BMS, supporting the increased use of Xience DES. Additionally, new-generation drug-eluting stents (DES) used in a more complex patient cohort to earlier comparative trials of DES and BMS demonstrated at least equivalent safety of the Xience DES to BMS in a real-world cohort, providing support for the continued and expanded use of this newer stent platform to improve patient outcomes.

## Figures and Tables

**Table 1 jcm-12-00280-t001:** Baseline Clinical Characteristics.

Characteristics *n* (%)	Overall *n* = 1500	BMS Only *n* = 500	Xience *n* = 1000	*p*-Value
Age, years, mean (SD)	68.4 (10.7)	71.5 (11.1)	66.9 (10.1)	<0.001
Male	1153 (76.9)	382 (76.4)	771 (77.1)	0.76
Diabetes	368 (24.6)	107 (21.5)	261 (26.2)	0.051
Hypertension	1094 (73.3)	369 (74.1)	725 (72.9)	0.61
Hypercholesterolaemia ^1^	1208 (83.5)	381 (79.4)	827 (85.6)	0.003
Family History of CAD	514 (37.8)	137 (29.3)	377 (42.2)	<0.001
Smoking (past or current)	822 (55.9)	275 (55.9)	547 (55.9)	0.99
BMI, kg/m^2^ ± SD	29.0 (4.9)	29.0 (5.0)	29.0 (4.9)	0.92
LVEF, mean ± SD	56.9 (10.4)	55.1 (11.2)	57.9 (9.8)	<0.001
Previous MI	311 (20.8)	83 (16.6)	228 (22.9)	0.005
Previous Peripheral Vascular Disease	120 (8.0)	50 (10.0)	70 (7.0)	0.046
Previous PCI	432 (28.9)	85 (17.1)	347 (34.7)	<0.001
Previous Cerebrovascular disease	114 (7.6)	51 (10.2)	63 (6.3)	0.008
Previous CABG	148 (9.9)	49 (9.8)	99 (9.9)	0.96
Renal impairment ^2^	82 (5.9)	38 (7.9)	44 (4.8)	0.019
Clinical presentation				
STEMI	143 (9.6)	100 (20.0)	43 (4.3)	<0.001
NSTEMI	360 (24.2)	145 (29.0)	215 (21.7)	<0.001
Unstable angina	244 (16.4)	73 (14.6)	171 (17.3)	ns
Elective	742 (49.8)	182 (36.4)	560 (56.6)	<0.001
Cardiogenic Shock	7 (0.5)	5 (1.0)	2 (0.2%	0.033

CAD—coronary artery disease; BMI—Body mass index; PCI—Percutaneous coronary intervention; CABG—Coronary artery bypass grafting; MI—Myocardial infarction; STEMI—ST-elevation myocardial infarction; NSTEMI—Non-ST-elevated myocardial infarction; SD—Standard deviation. ^1^ Hypercholesterolaemia is defined as either (a) Cholesterol level >5.2 and/or (b) Receiving medication. ^2^ Renal impairment is defined as either (a) Sr. Creatinine >2 mg/dl and/or (b) having renal failure/receiving treatment.

**Table 2 jcm-12-00280-t002:** Lesion & Procedural Characteristics.

Characteristics *n* (%)	Overall *n* = 1500	BMS *n* = 500	Xience *n* = 1000	*p*-Value
Lesions/procedure, mean (SD)	1.3 (0.6)	1.2 (0.5)	1.4 (0.7)	<0.001
Access site				<0.001
Radial	460 (30.7)	117 (23.4)	343 (34.3)	
Femoral	1036 (69.1)	383 (76.6)	653 (65.4)	
Brachial	3 (0.2)	0 (0.0)	3 (0.3)	
Lesion Type				0.002
De novo	1425 (95.0)	488 (97.6)	937 (93.7)	
Restenosis	2 (0.1)	0 (0.0)	2 (0.2)	
In-stent restenosis	62 (4.1)	7 (1.4)	55 (5.5)	
Other	10 (0.7)	4 (0.8)	6 (0.6)	
ACC/AHA Morphology				<0.001
A	157 (10.6)	70 (14.1)	87 (8.8)	
B1	637 (42.8)	229 (46.0)	408 (41.3)	
B2 or C	693 (46.6)	199 (40.0)	494 (49.9)	
Target vessel				<0.001
RCA	496 (33.3)	202 (40.6)	294 (29.7)	
LMCA	14 (0.9)	5 (1.0)	9 (0.9)	
LAD	628 (42.2)	161 (32.3)	467 (47.1)	
LCx	298 (20.0)	105 (21.1)	193 (19.5)	
Bypass Graft	53 (3.6)	25 (5.0)	28 (2.8)	
Chronic total occlusion	27 (1.8)	5 (1.0)	22 (2.2)	0.096
Multi-vessel disease	526 (35.1)	170 (34.0)	356 (35.6)	0.58
Bifurcation lesion	138 (9.3)	24 (4.8)	114 (11.5)	<0.001
Lesion success	1481 (99.5)	496 (99.6)	985 (99.5)	0.78
Stents/procedure, mean (SD)	1.4 (0.7)	1.3 (0.7)	1.5 (0.8)	<0.001
Stent length, mean (SD)	18.1 (5.7)	17.0 (4.7)	18.7 (6.1)	<0.001
Stent length >20 mm	387 (25.8)	85 (17.0)	302 (30.2)	<0.001
Stent diameter, mean (SD)	3.1 (0.5)	3.3 (0.6)	2.9 (0.4)	<0.001
Vessel ≤ 2.5 mm	316 (21.1)	64 (12.8)	252 (25.2)	<0.001
IIb/IIIa use during procedure	103 (6.9)	47 (9.4)	56 (5.6)	0.006

RCA—Right coronary artery; LMCA—Left main coronary artery; LAD—Left anterior descending artery; LCX—Left circumflex artery; BMS—Bare metal stent; SD—Standard deviation.

**Table 3 jcm-12-00280-t003:** Clinical Outcomes in Hospital, at 30 days and 12 months.

Outcome *n* (%)	Overall *n* = 1500	BMS *n* = 500	Xience *n* = 1000	*p*-Value
In-hospital				
Emergency PCI	3 (0.2%)	2 (0.4%)	1 (0.1%)	0.22
CABG	0	0	0	
Major Bleeding *	25 (1.7%)	7 (1.4%)	18 (1.8%)	0.56
Myocardial infarction	44 (3.0%)	12 (2.4%)	32 (3.2%)	0.38
Death	6 (0.4%)	4 (0.8%)	2 (0.2%)	0.083
30-Day Follow-up				
Eligible procedures	1494	496	998	
Followed up	1494 (100.0%)	496 (100.0%)	998 (100.0%)	
Death	3 (0.2%)	3 (0.6%)	0 (0.0%)	0.014
Myocardial infarction	0	0	0	
TVR	4 (0.3%)	2 (0.4%)	2 (0.2%)	0.47
TLR	4 (0.3%)	2 (0.4%)	2 (0.2%)	0.47
MACE (composite)	7 (0.5%)	5 (1.0%)	2 (0.2%)	0.031
Procedures lead to any readmission	103 (6.9%)	52 (10.5%)	51 (5.1%)	<0.001
Any unplanned readmission	26 (1.7%)	10 (2.0%)	16 (1.6%)	0.57
12-month Follow-up				
Eligible procedures	1491	493	998	
Followed up	1477 (99.1%)	488 (99.0%)	989 (99.1%)	
Death	25 (1.7%)	14 (2.9%)	11 (1.1%)	0.014
Myocardial infarction	9 (0.6%)	4 (0.8%)	5 (0.5%)	0.47
TVR	14 (0.9%)	3 (0.6%)	11 (1.1%)	0.35
TLR	16 (1.1%)	8 (1.6%)	8 (0.8%)	0.15
MACE (composite)	47 (3.2%)	20 (4.1%)	27 (2.7%)	0.16
Procedure lead to any readmission	286 (19.4%)	119 (24.4%)	167 (16.9%)	<0.001
Any unplanned readmission	110 (7.4%)	37 (7.6%)	73 (7.4%)	0.89

* Major bleeding = BARC 3, 4 or 5; PCI—Percutaneous coronary intervention; CABG—Coronary artery bypass grafting; TVR—Target vessel revascularization; TLR—Target lesion revascularization; MACE—Major adverse cardiovascular events.

**Table 4 jcm-12-00280-t004:** 1-year Outcomes for BMS vs, Xience DES by Acuity.

Outcome		Overall			Elective			Acute		
	Overall	BMS	Xience	*p*	BMS	Xience	*p*	BMS	Xience	*p*
Death	28 (1.9%)	17 (3.5%)	11 (1.1%)	0.002	5 (2.8%)	3 (0.5%)	0.01	12 (3.9%)	8 (1.9%)	0.11
OR* Xience vs. BMS	0.30 (0.10–0.91) *p* = 0.03				0.16 (0.03–0.92) *p* = 0.04			0.40 (0.11–1.47) *p* = 0.17		
MACE	51 (3.5%)	22 (4.5%)	29 (2.9%)	0.12	8 (4.5%)	10 (1.8%)	0.04	14 (4.5%)	19 (4.5%)	0.99
OR* Xience vs. BMS	0.70 (0.33–1.48) *p* = 0.35				0.63 (0.20–1.98) *p* = 0.43			0.99 (0.43–2.30) *p* = 0.98		
Unplanned readmission	131 (8.9%)	44 (9.0)	87 (8.8%)	0.89	16 (8.9%)	40 (7.2%)	0.44	28 (9.1%)	47 (11.2%)	0.35
OR* Xience vs. BMS	0.88 (0.56–1.38) *p* = 0.58				0.77 (0.36–1.66) *p* = 0.51			1.07 (0.61–1.89) *p* = 0.81		

* OR adjusted Odds ratio.

**Table 5 jcm-12-00280-t005:** Discharge and 1-Year Medication Compliance rates.

Drug Therapy *n* (%)	Overall	BMS	Xience	*p*-Value
	1500	500	1000	
Discharged alive, *n*	1494	496	998	
Aspirin	1463 (98.3%)	484 (97.6%)	979 (98.6%)	0.16
Clopidogrel/Prasugrel/Ticagrelor	1472 (98.5%)	487 (98.2%)	985 (98.7%)	0.44
Statin	1407 (94.6%)	468 (94.4%)	939 (94.8%)	0.75
B-blocker	849 (57.2%)	310 (62.5%)	539 (54.5%)	0.003
ACE/ARB	1023 (68.5%)	330 (66.5%)	693 (69.4%)	0.25
30-day outcomes	1494	496	998	
Aspirin	1373 (96.4%)	432 (93.7%)	941 (97.7%)	<0.001
Clopidogrel/Prasugrel/Ticagrelor	1392 (95.8%)	438 (91.6%)	954 (97.8%)	<0.001
Any Antiplatelet	1424 (98.0%)	460 (96.2%)	964 (98.9%)	<0.001
Statin	1346 (94.7%)	431 (93.7%)	915 (95.1%)	0.27
B-blocker	766 (53.8%)	279 (60.5%)	487 (50.6%)	<0.001
ACE/ARB	959 (66.0%)	303 (63.4%)	656 (67.3%)	0.14
12-month outcomes	1477	488	989	
Aspirin	1228 (91.6%)	363 (85.4%)	865 (94.4%)	<0.001
Clopidogrel/Prasugrel/Ticagrelor	903 (64.8%)	197 (43.7%)	706 (74.9%)	<0.001
Any Antiplatelet	1300 (93.3%)	394 (87.4%)	906 (96.2%)	<0.001
Statin	1247 (93.3%)	394 (93.1%)	853 (93.4%)	0.85
B-blocker	629 (47.2%)	223 (52.8%)	406 (44.6%)	0.005
ACE/ARB	902 (64.8%)	264 (58.5%)	638 (67.7%)	<0.001

B-blocker—Beta blocker; ACE—Angiotensin converting enzyme inhibitors; ARB—Angiotensin II receptor blockers.

## Data Availability

The data underlying this article will be shared on reasonable request to the corresponding author.

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
