# Peer review of "Long-Term Outcomes of Contemporary Percutaneous Coronary Intervention with the Xience Drug-Eluting Stent: Results from a Multicentre Australian Registry"

_jcm, 2022, doi:10.3390/jcm12010280_

Round 1
Reviewer 1 Report
This study represents the first registry of national scale reporting outcomes following PCI with Xience DES in Australia. It provides a unique perspective on contemporary prac-tice and outcomes of PCI in Australia, and a large-scale report regarding PCI performance in Australian private hospitals. In my opinion, it might be interest to the readers of JCM. However, it would be great if the authors could provide more discussion regarding the difference of outcomes following PCI with Xience DES in Australia than other areas.
Author Response
This study represents the first registry of national scale reporting outcomes following PCI with Xience DES in Australia. It provides a unique perspective on contemporary prac-tice and outcomes of PCI in Australia, and a large-scale report regarding PCI performance in Australian private hospitals. In my opinion, it might be interest to the readers of JCM. However, it would be great if the authors could provide more discussion regarding the difference of outcomes following PCI with Xience DES in Australia than other areas.
Thank you for your suggestion and we have revised and expanded the introduction and discussion (please see highlighted text) to provide comparison with other countries.
Reviewer 2 Report
Manuscript ID: jcm-2080879
Title: Long-term Outcomes of contemporary Percutaneous Coronary Intervention with the Xience Drug-Eluting Stent: Results from a Multicentre Australian Registry
Comments to the authors
Manuscript is well written.
Elective cases and ACS cases should be considered separately because the mechanism of onset is different and the risk of subsequent thrombosis is different.
Since it is related to TVR and TLR, please describe it if you know the usage rate of intravascular imaging.
Author Response
Elective cases and ACS cases should be considered separately because the mechanism of onset is different and the risk of subsequent thrombosis is different.
Thank you for your comments and suggestions and we have added a new section in results (highlighted text) to describe characteristics and outcomes of Xience vs BMS patients stratified by Acute vs Elective presentations, with table for adjusted odds ratio. We have also included Supplementary Tables 1 and 2 to describe patient and procedural characteristics based on acuity presentation.
Since it is related to TVR and TLR, please describe it if you know the usage rate of intravascular imaging.
Unfortunately, we do not have data on the usage rate of intravascular imaging, although both IVUS and OCT are utilised across the network.
Round 2
Reviewer 2 Report
The manuscript has been revised well. I think this manuscript will be acceptable.